# Prevalence and Possible Etiological Factors of Molar-Incisor Hypomineralization (MIH) in Population of Silesian Children in Poland: A Pilot Retrospective Cohort Study

**DOI:** 10.3390/ijerph19148697

**Published:** 2022-07-17

**Authors:** Danuta Ilczuk-Rypuła, Marzena Zalewska, Daria Pietraszewska, Anna Dybek, Aleksandra Nitecka-Buchta, Lidia Postek-Stefańska

**Affiliations:** 1Department of Pediatric Dentistry, Faculty of Medical Sciences in Zabrze, Medical University of Silesia, 40-055 Katowice, Poland; dpietraszewska@sum.edu.pl (D.P.); annadybek20@gmail.com (A.D.); swrzab@sum.edu.pl (L.P.-S.); 2Department of Medical and Molecular Biology, Faculty of Medical Sciences in Zabrze, Medical University of Silesia, 40-055 Katowice, Poland; mzalewska@sum.edu.pl; 3Department of Temporomandibular Disorders, Faculty of Medical Sciences in Zabrze, Medical University of Silesia, 40-055 Katowice, Poland; aleksandra.nitecka@sum.edu.pl

**Keywords:** molar-incisor hypomineralization (MIH), children, frequency of prevalence, environmental etiological factors, pediatric dentistry

## Abstract

(1) Background: This pilot retrospective cohort research study regarded the frequency of the prevalence of molar-incisor hypomineralization (MIH) in a population of Silesian children in Poland. The evaluation of the potential environmental etiological factors was performed and the correlation between the environmental factors and MIH was analyzed. (2) Methods: A total of 613 children were randomly enrolled in the pilot study (9.0 years ± 1.9). A survey was carried out with mothers regarding the potential exposure to environmental etiological factors of MIH in their children. The patients’ clinical assessments were carried out in the dental unit. (3) Results: The frequency of prevalence of MIH in the population of Silesian children was established at the level of 6.2% (*p* < 0.05). MIH symptoms were significantly associated with otitis in early childhood (OR = 2.50), atopic dermatitis (OR = 5.69), and premature delivery before 38 weeks of pregnancy (OR = 2.88). (4) Conclusions: MIH was observed in 6.2% of the population of Silesian children, and there was a relationship between environmental risk factors such as otitis, atopic dermatitis, premature birth, and MIH expression. Therefore, further research is needed to determine the influence of pre-, peri-, postnatal, and prophylactic factors on the frequency and severity of MIH symptoms in children.

## 1. Introduction

Molar-incisor hypomineralization (MIH) was defined by Weerheijm in 2001 as a qualitative enamel defect impacting at least one first permanent molar tooth (FPM) and very often permanent incisors (PI) [1]. Due to the relevancy of MIH medical condition, the analysis of etiological factors is carried out by many researchers. Knowledge in this field would help in the implementation of prevention procedures. However, the multifactorial and ambiguous etiology of MIH makes it difficult to determine the relation.

Amongst the etiological factors, the environmental factors in the prenatal, perinatal, and postnatal periods can be distinguished. Fatturi, in a review of systemic exposure in MIH patients, observed that the children of mothers who had suffered from health problems during pregnancy had a 40% greater chance of developing MIH compared to children whose mothers did not suffer from any health problems during pregnancy [2].

In the prenatal period, the factors that can potentially affect ameloblasts’ development are viral infections of a pregnant mother and episodes/periods of high fever, hypertension, diabetes, or medication use in pregnant women [3,4]. The perinatal period and its related complications during labor are premature delivery, cesarean section, or low birth weight, which may also be attributed as potential MIH factors. These conditions contribute to the low oxygen supply during the process of amelogenesis [5]. In the postnatal period, a high exposure to the relevant environmental factors can be a significant etiological factor for MIH expression. These factors mainly include the pollution of the environment by dioxins, the previous medical history of the child, and the intake of prescribed medications [6]. The correlation between the development of MIH and the type and time of baby feeding was observed [7]. Some scientists attribute the influence of the level of vitamin D and the pollution of the environment to the development of MIH [8]. Scientific reports indicate that the development of MIH may be influenced by genetic factors or the additive and synergistic influence of many etiological factors of different origins [9,10]. Kühnisch et al. [10], in a genome-wide association study (GWAS) of German children, investigated the relationship between molar-incisor hypomineralization (MIH) and possible genetic loci. The main finding of their study was that SCUBE1 (rs13058467) is a possible MIH-related gene locus [10]. In their research study, Jeremias et al. confirmed the influence of genetic factors on the development of MIH [11]. This family association study performed on a population of Brazilian families proved that changes in amelogenesis-related genes such as the FAM83H gene (rs7821494), AMBN gene (rs34367704), BMP2 gene (rs3789334), BMP7 gene (rs6099486), BMP4 gene (rs762642), ENAM gene (rs7664896), MMP20 gene (rs1711399, rs1711423), DLX3 gene (rs2278163), FGFR1 gene (rs6996321), and AMELX gene (rs5979395) are associated with susceptibility to the development of MIH [11]. Additionally, Teixeira et al., examining pairs of twins, found that there is a greater concordance of MIH among pairs of monozygotic twins compared to dizygotic twins, suggesting the genetic etiology of this disease [12]. However, this possibility is difficult to define unequivocally as there are many indications that the etiology of MIH is multifactorial and ambiguous.

The frequency of prevalence of MIH varies and differs in percentage from 2.4% to 44%, depending on the research [13,14,15,16,17]. In Denmark, the problem of molar-incisor hypomineralization is more common than caries on the occlusal surface of these teeth [18]. Clinically, depending on the severity of this syndrome, the changes can vary from white-cream or yellow-brown opacities to large cavities in the hard tissues of the teeth, which contributes to their rapid destruction. The decreased mineralization and increased porosity of the enamel that characterize MIH cause the weakening of the enamel’s structure [19].

In a study by Raposo et al., it was shown that there is a correlation between MIH and the incidence of hypersensitivity in the affected teeth [20]. This can negatively affect the cooperation with pediatric dentistry patients. For an accurate assessment of the clinical view, the MIH Treatment Need Index (MIH-TNI) has been developed. This is a tool for the planning and implementation of the optimal treatment plan depending on the clinical situation [17]. The MIH-TNI classification is based on the two most important clinical symptoms: tooth destruction and hypersensitivity. Table 1 presents the values of the MIH-TNI index [17].

The aim of this pilot, cohort, and retrospective study was the assessment of MIH frequency/prevalence and MIH intensity in a population of Silesian children in Poland. The authors determined the potential risk of MIH development under different environmental factors affecting the population of Silesian children.

## 2. Materials and Methods

The pilot study was performed in children (*n* = 613) attending the Developmental Age Clinic of the Academic Center of Dentistry and Specialized Medicine in Bytom between 1 January 2019 and 1 March 2022. Patients were randomly enrolled in the study. Parents scheduled appointments for their children with respect to a routine dental control or conservative dental treatment visit. Patients were registered with D.I.R, D.P., or L.P.-S for the control visit or the routine conservative dentistry treatment. Medical anamnesis data obtained from the interview with the patient’s guardian or parent were collected in the form of a questionnaire, supplemented by the attending physician in cooperation with the child’s guardian.

Patient’s clinical intraoral examination was carried out on the dental unit with intraoral mirror, probe, and dental light. Examination was performed by members of the research team: D.I.R., D.P. and L.P.-S. The final MIH diagnosis was confirmed by at least two investigators independently—the pediatric dentists, who were calibrated to diagnose MIH. Based on the criteria of severity of MIH according to the EAPD, two main forms were observed: mild MIH and severe MIH. The criteria used for the diagnosis of MIH were in accordance with the criteria introduced during the European Academy of Pediatric Dentistry (EAPD) meeting, which took place in Athens in 2003 [21], and the protocol of the research with the inclusion/exclusion criteria (Table 2). The different levels of EAPD classification are presented in the Table 3 [4].

Additionally, inter-examiner reproducibility was calculated and was found to be high in all examined parameters (kappa = 0.92–0.95). The study protocol was approved by the Bioethical Committee of the Medical University of Silesia in Katowice, Poland (PCN/0022/KB1/108/19).

After obtaining parents’ permission and completion of the entry form, the patients’ age, gender, and the severity of their MIH were assessed, in accordance with the EAPD diagnostic criteria [21] as well as the MIH-TNI indicator [17]. After collecting questionnaire information, the potential exposure to the etiological environmental MIH factors was established. The questionnaire was filled in during a face-to-face interview with the child’s mother and the data contained in the child’s health book. The questionnaire consisted of questions related to the following issues:the demographic data;the maternal health during pregnancy, folic acid, vitamin supplements, and consumption of stimulants during pregnancy;the time and the type of delivery of the child;the child’s birth weight;the breastfeeding time;the medical history of the child in first three years of life (chickenpox, otitis, bronchitis, pneumonia, atopic dermatitis, asthma, fever above 39 °C, and intake of corticosteroids).

Collected data were saved in MS Excel file. A statistical analysis of the obtained results was carried out with use of Statistica Software 12.6 (Stat Soft Inc., Tulsa, OK, USA) in an impersonal form. Personal data protection has been applied.

A statistical *t*-test and Mann–Whitney U-test were used to analyze the distribution of data. The odds ratio (OR) and the relative risk (RR) with a 95% confidence interval (CI) were calculated to find the risk of MIH in conjunction with potential prenatal, perinatal, and postnatal etiological factors. The statistical significance was assumed with *p* < 0.05.

## 3. Results

The average age of the population of Silesian children in this research was 9.1 years (SD = ±1.9 years). In the population of 613 patients, the average age of mothers was 28.8 years (SD = ±3.8 years) and the average age of fathers was 30.6 years (SD = ±4.7 years). A total of 54.54% of mothers had higher education and the figure was 37.83% for fathers. In the analyzed population, 53.13% of children were the first-born children of their respective mothers. A total of 85.48% of children in the analyzed population were born via a natural birth.

In the population of Silesian children enrolled in the study (*n* = 613), symptoms of molar-incisor hypomineralization (MIH) were found in 38 patients (MIHn = 38). The frequency of prevalence of MIH in the analyzed population was 6.2%. The average age of children with reported MIH was 9.1 years (SD = ±1.7 years). In the MIH group, there were 17 girls (44.73% female) and 21 boys (55.27% male). In the non-MIH group, there were 575 children (non-MIHn = 575) comprising 302 girls (52.52% female) and 273 boys (47.48% male). The average age of the children without MIH was 9.0 years (SD = ±1.8 years).

Based on the criteria of the severity of MIH according to the EAPD, mild MIH (mMIH) was diagnosed in 13 children (mMIHn = 34.21%) and severe MIH (sevMIH) was diagnosed in 25 children (sevMIHn = 65.79%) (Table 4).

In 613 children assessed in accordance with the Wurzburg concept, MIH-TNI 2C (without hypersensitivity of the teeth and with a cavity comprising over 2/3 of the tooth surface) was diagnosed in 34.21% of them (MIH-TNI 2C). The largest percentage of MIH-TNI 2C was found in females (47.07%) (Table 5).

A total of 23.68% of children from the MIH group showed MIH lesions only in the first permanent molars. In this group, four first permanent molars and two permanent incisors were affected in 21.05% of the children. The detailed distribution of the children through a combination of the affected teeth in the MIH research group together with all the affected permanent teeth with MIH symptoms is presented in Table 6.

Moreover, 71.43% of the assessed patients were diagnosed with MIH without hypersensitivity (MIH-TNI 1 and MIH-TNI 2) with different degrees of changes.

The distribution of the potential etiological factors in children in the MIH group and the non-MIH group is presented in Figure 1 and Table 7. Otitis in early childhood (until the third year of life), atopic dermatitis, and a preterm birth before 38 weeks of pregnancy were significantly associated with MIH. Breast-feeding showed a slight protective effect on the development of MIH in the trend line, but it did not show a statistically significant relationship. Other potential etiological factors, such as prenatal, perinatal, and postnatal factors, were not significantly associated with MIH.

## 4. Discussion

The aim of this pilot research study was to assess the frequency of prevalence of molar-incisor hypomineralization amongst the population of Silesian children. The prevalence of MIH measured in the analyzed group was 6.20% and it falls into the wide spectrum of values noted in the research carried out around the world—values of 2–44% [13,17]. Moreover, this value is comparable to the frequency received in the research carried out on the population of children of the northern regions of Poland—6.43% [22]. This research is one of very few carried out on a population of Polish pediatric patients. In addition, similar results were achieved by Preusser et al., who analyzed the population of children in Germany—5.9% [23] and Buchgraber et al. amongst children in Austria—7% [24]. The frequency of prevalence of MIH was lower than in other parts of Europe. Lygidakis et al. [25] during research carried out in Greece assessed the frequency of prevalence of MIH at the level of 10.2%. Amend et al. had received a similar frequency for children in rural parts of Germany at 9.4% and much higher in urban areas at 17.4% [26]. A higher frequency of prevalence of MIH in Europe was detected in the Netherlands—14.3% [27], Italy—13.7%, [28] and Bosnia and Herzegovina—12.3% [29]. A similar disproportion of results can be observed outside of Europe. Saber et al., in their research in Egypt, achieved an MIH level in their pediatric population of 2.3% [30]. Davenport et al., in pilot research conducted in the USA, assessed the frequency of prevalence of MIH at 9.6% [31]. Moreover, a significantly higher prevalence of MIH was observed by Wainuiomata (14.9%) [32] in India—13.12% [33], Dubai—27.2% [34], Saudi Arabia—15.2% [35], and Lebanon—26.7 % [36]. To establish the accurate frequency of prevalence of MIH in Poland, thorough research must be carried out in Upper Silesia as well as other parts of Poland.

The association between the prevalence and severity of MIH and the sex of the children was analyzed; however, no statistically significant correlation was found. The results achieved herein are in line with the results achieved by other scientists [22,37,38]. Unlike other studies, 55.27% of children diagnosed with MIH were male in this study [39,40] Similar to the results achieved by Lygidakis et al. [37], in the research herein the majority of the children assessed had MIH present in all four permanent molars. However, in comparison to the research carried out in northern Poland [22], the majority of children assessed had been diagnosed with MIH characterized as severe. Most of the female children (52.95%) diagnosed with MIH had severe lesions (covering more than 2/3 of the tooth surface—MIH-TNI 2c and 4c). Chawla et al. [41] suggests that FPMs affected by MIH in female children may be more advanced in eruption than in male children, exposing enamel hypomineralization to masticatory forces and promoting the earlier onset of the post-eruptive enamel breakdown (PEB). This is probably because the dental age may be more advanced in women than in men at the ages of 5 to 16 years [41].

The statistical analysis proved a positive relation between the prevalence of MIH and potential etiological factors, such as otitis in early childhood (up to the 3rd year of age), atopic dermatitis, and a premature childbirth before 38 weeks of pregnancy. The results are consistent with the results obtained by Garot et al. [42], who, based on meta-analysis, noticed that the development of MIH is related to several systemic and genetic and/or epigenetic factors acting synergistically or additively, revealing a multifactorial etiological model. Among the etiological factors that more often increase the probability of MIH induction, these factors included prematurity, cesarean delivery, measles, urinary tract infection, otitis media, gastric disorders, bronchitis, kidney diseases, pneumonia, and asthma [42]. Mishra and Pandey also agree with the results achieved and noted a statistically significant correlation between an ear infection in early childhood, vitamin A deficiency, and MIH [43]. Similar to the present study, Koruyucu et al. [44] showed a significant association between MIH and ear infection during early childhood and in premature births. Moreover, they found that the potential etiological factors of MIH may also include environmental factors such as complications during the mother’s pregnancy, the frequency of diarrhea, asthma, a frequent high fever, chickenpox, and parotitis [44]. In a manner similar to this research, Almuallem et al. pointed out that childhood illnesses (ear infection, respiratory distress, and tonsillitis) during the first three years of life were significantly associated with MIH [35]. This pilot study is also in agreement with those results recorded by Giuca et al. [45], who observed a positive association between molar-incisor hypomineralization and the ear, nose, and throat disorders during early childhood. Silva et al. [46] concludes that early childhood diseases, in conjunction with a high body temperature, could be an etiological factor in MIH. However, they also suggested further prospective research with a better control of the factors disrupting amelogenesis to better understand the real etiology of MIH [46]. Prescribed medication, especially antibiotics, are also researched as a potential etiological factor of MIH. Additionally, some scientists have noticed a correlation between taking antibiotics and the prevalence of MIH [47,48,49]. On the other hand, in a systematic review, prescribed medicine was not significantly connected with MIH [50].

Allazzam et al. and Alhowaish et al. [5,51] disagreed with the results we obtained and did not note any significant correlation between the perinatal and prenatal factors and MIH. However, Alhowaish et al. [51] were the first to indicate that hepatitis can be a potential perinatal etiological factor enabling the occurrence of MIH in children.

Some of the limitations of this pilot study must be pointed out. First, it is not possible to generalize the observations for the whole population of children in the Upper Silesia region or in Poland itself. Further research is needed for the analysis of the possible etiological factors and the risk of MIH occurrence. The authors of this pilot study consider further research, including participants from other centers and patients from different regions of Poland. In addition, the current authors’ cooperation with other researchers was significantly hampered by the COVID-19 pandemic. The authors hope to continue their cooperative research to a greater extent in the future.

## 5. Conclusions

Based on the results of the research, the conclusions below were formulated.
The prevalence of MIH in the population of Silesian children in Poland was 6.2%.The population of Silesian children diagnosed with MIH (*n* = 38) showed the features of severe MIH in 65.79%; severe MIH occurred in 47.07% of girls diagnosed with MIH.Etiological factors: otitis in early childhood, atopic dermatitis, and premature childbirth showed statistically significant correlations with MIH prevalence.The study’s results fall within the range of other countries’ MIH levels and are similar to those achieved in northern Poland.Currently, MIH is a worrying problem in pediatric dentistry and further research is needed to assess its potential etiological factors and prevention.

## Figures and Tables

**Figure 1 ijerph-19-08697-f001:**
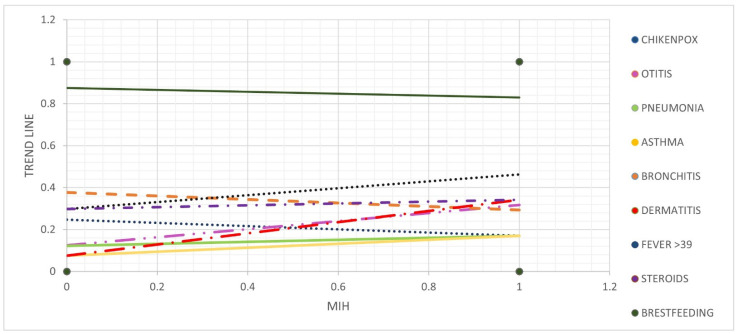
Etiological factors influencing MIH with a trend line.

**Table 1 ijerph-19-08697-t001:** The MIH Treatment Need Index.

Index	Definition
Index 0	No MIH, clinically free of MIH
Index 1	MIH without hypersensitivity, without defect
Index 2	MIH without hypersensitivity, with defect
2a	<1/3 defect extension
2b	>1/3 <2/3 defect extension
2c	>2/3 defect extension or/and defect close to the pulp or extraction or atypical restoration
Index 3	MIH with hypersensitivity, without defect
Index 4	MIH with hypersensitivity, with defect
4a	<1/3 defect extension
4b	>1/3 <2/3 defect extension
4c	>2/3 defect extension or/and defect close to the pulp or extraction or atypical restoration

**Table 2 ijerph-19-08697-t002:** The inclusion and exclusion criteria of the pilot study.

The Inclusion Criteria	The Exclusion Criteria
Age 8–12 years	Actual orthodontic treatment
First permanent molars had erupted	Developmental defects of teeth: amelogenesis imperfecta enamel hypoplasia fluorosis tetracycline staining
Minimum of six permanent incisors had erupted	Children with genetic disorders
Residency in Silesian Voivodeship, Poland	Children with birth defects
	MIH symptoms only limited to the incisors
	Parents’ refusal to consent to participate in the study

**Table 3 ijerph-19-08697-t003:** The severity levels according to the EAPD classification.

The Severity Level	Signs and Symptoms
MILD	Demarcated enamel opacities without enamel breakdown Induced sensitivity to external stimuli, e.g., air/water, but not brushing Mild aesthetic concerns regarding discoloration of the incisors
SEVERE	Demarcated enamel opacities with breakdown and caries Spontaneous and persistent hypersensitivity affecting function, e.g., brushing, mastication, etc. Strong aesthetic concerns that may have socio-psychological impact

**Table 4 ijerph-19-08697-t004:** Severity of MIH according to the EAPD.

Severity of MIH	Female	Male	Grand Total	Grand Total (%)
Mild	5	7	13	34.21%
Severe	12	14	25	65.79%

**Table 5 ijerph-19-08697-t005:** Severity of MIH according to the Wurzburg concept (MIH-TNI).

Severity of MIH	Female	Male	Grand Total	Female (%)	Male (%)	Grand Total (%)
1	3	5	8	17.65%	23.81%	21.05%
2A	1	0	1	5.88 %	0.00%	2.63%
2B	1	5	6	5.88%	23.81%	15.80%
2C	8	5	13	47.07%	23.81%	34.21%
3	0	2	2	0.00%	9.52%	5.26%
4A	2	0	2	11.76%	0.00%	5.26%
4B	1	1	2	5.88%	4.76%	5.26%
4C	1	3	4	5.88%	14.29%	10.53%
Grand Total	17	21	38	100%	100%	100%

**Table 6 ijerph-19-08697-t006:** The distribution of affected teeth in a group of Silesian children with MIH.

Teeth Affected									
FPMs with MIH	Alone	+1 PI	+2 PI	+3 PI	+4 PI	+5 PI	+6 PI	+7 PI	+8 PI
**0 molar**	0	0	0	0	0	0	0	0	0
**1 molar**	2	0	3	0	0	0	0	0	0
**2 molars**	0	0	1	1	0	0	1	0	0
**3 molars**	4	1	1	1	0	0	0	0	0
**4 molars**	3	3	**8**	3	2	0	2	2	0

**FPM**—first permanent molar; **PI**—permanent incisor.

**Table 7 ijerph-19-08697-t007:** The distribution of potential etiological environmental factors in the MIH and non-MIH groups.

	MIH	Non-MIH	RR	95% CI		OR	95% CI
Maternal cigarette smoking during pregnancy	yes	6	43	2.15	[0.95–4.90]	2.31	[0.92–5.83]
	no	32	532				
Maternal alcohol intake during pregnancy	yes	2	14	2.02	[0.53–7.70]	2.17	[0.48–9.88]
	no	36	561				
Maternal folic acid supplementation during pregnancy	yes	32	518	0.62	[0.27–1.42]	0.59	[0.24–1.48]
	no	6	58				
Vitamin supplementation during pregnancy	yes	22	316	1.12	[0.60–2.08]	1.13	[0.58–2.19]
	no	16	259				
Preterm childbirth before 38 weeks of pregnancy	yes	5	29	2.60 *	[1.08–6.23]	2.88 *	[1.05–7.92]
	no	33	546				
Chickenpox before 3rd year of life of child	yes	6	144	0.58	[0.25–1.36]	0.56	[0.23–1.37]
	no	32	431				
Otitis before 3rd year of life of child	yes	10	72	2.32 *****	[1.17–4.59]	2.50 *	[1.17–5.36]
	no	28	503				
Bronchitis before 3rd year of life of child	yes	12	216	0.78	[0.40–1.52]	0.77	[0.38–1.56]
	no	26	359				
Pneumonia before 3rd year of life of child	yes	7	72	1.53	[0.70–3.35]	1.58	[0.67–3.72]
	no	31	503				
Asthma before 3rd year of life of child	yes	6	43	2.15	[0.95–4.90]	2.31	[0.92–5.83]
	no	32	532				
Atopic dermatitis before 3rd year of life of child	yes	12	43	4.67 *	[2.50–8.73]	5.69 *	[2.69–12.06]
	no	26	532				
Episodes of fever above 39 °C before 3rd year of life of child	yes	17	173	1.81	[0.98–3.35]	1.89	[0.97–3.67]
	no	21	403				
Corticosteroid therapy before 3rd year of life of child	yes	14	173	1.33	[0.71–2.52]	1.36	[0.69–2.69]
	no	24	403				
Breastfeeding up to 12 months of age	yes	34	503	1.20	[0.44–3.29]	1.21	[0.42–3.52]
	no	4	72				

***** indicates STATISTICALLY SIGNIFICANT. RR—The Relative Risk; CI—The Confidence Interval; OR—The Odds Ratio.

## Data Availability

Data supporting the findings of the present study can be requested from authors.

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
