# Peer review of "Prevalence and Possible Etiological Factors of Molar-Incisor Hypomineralization (MIH) in Population of Silesian Children in Poland: A Pilot Retrospective Cohort Study"

_ijerph, 2022, doi:10.3390/ijerph19148697_

Round 1

Reviewer 1 Report

1. English language must be corrected

2. More detailed statistical description would be fine.

3. Statistical analyses is mostly descriptive. Did authors evaluate any correlations? If not, why?

4. How authors received and evaluated the parental information? Questionnaire? 

Author Response

On behalf of the authors of the work, thank you for reading our article and for critically evaluating it. We appreciate the very detailed and extremely pertinent comments that allowed us to improve our manuscript. The authors of the study made every effort to improve the text in accordance with the guidelines of the Reviewers.

  • As suggested by the Reviewer, the statistical analysis of the obtained results has been improved. To the descriptive part of the statistics, statistical analysis of the study group and the control group has been added. The relative risk and the odds ratio were determined.
  • After  obtaining parent's permission, the questionnaire was filled in during a face-to-face interview with the child's mother and the data contained in the child's health book. This information has been added in the manuscript in the Material and Methods' chapter.
  • Linguistic errors throughout the text have been analyzed and corrected.

On behalf of the Authors, I apopgize for omissions and thank you for pointing out the errors and weaknesses of this article. At the same time, we hope that the implemented amendments will improve the quality of article and make it possible to publish it.

Reviewer 2 Report

Dear authors

The prevalence data are ok, but the associative analysis is rather awkward, since only mild and severe cases were compared at rather low numbers of cases (35), but no (healthy) control was included. Thus, it is not asthonishing that nothing meaningful can be deduced from the correlation data. A control group should have been added, maybe this can still be done? The conclusions based on the current correlations should be at least modified or, better, these data should be deleted.

Author Response

On behalf of the Authors of the article, thank you for your valuable comments.
The Authors made every effort to improve the text in line with the Reviewers' guidelines.

  • As suggested by the Reviewer, the statistical analysis of the obtained results has been improved. The study group and the control group were analyzed statistically. The relative risk and odds ratios were determined.

  • Conclusions have been corrected and completed in light of the results obtained.

  • Linguistic errors throughout the text have been analyzed and corrected.

On behalf of all the Authors, I would like to ask you to re-evaluate the improved manuscript. I sincerely hope that all comments have been correctly interpreted and applied in the article.

Reviewer 3 Report

Please attached PDF file for my comments.

Author Response

On behalf of the Authors of the article, thank you for reading our manuscript and for critically evaluating it. We appreciate the very detailed and extremely pertinent comments that allowed us to improve our manuscript. The Authors of the study made every effort to improve the text in accordance with the guidelines of the Reviewers.

  • As suggested by the Reviewer, the part, concerning the etiology of MIH, was moved to the first paragraph of the Introduction and extended.
  • The statistical analysis of the obtained results was improved. To the descriptive part of the statistics, the statistical analysis of the study group and the control group was added. The relative risk and the odds ratio were determined.
  • The Results and the Abstract have been corrected.
  • As suggested by the Reviewer, the type of the study was specified, both in the title of the article, as well as in the Abstract and in the Introduction.
  • Linguistic errors throughout the text have been analyzed and corrected.

On behalf of the Authors, I apologize for omissions and thank you for pointing out the errors and weaknesses of this article. At the same time, we hope that the implemented amendments will improve the quality of the article and make it possible to publish it.

Round 2

Reviewer 2 Report

Tha authors have considerably improved the manu.